# Development of clinical decision rules for traumatic intracranial injuries in patients with mild traumatic brain injury in a developing country

Tanat Vaniyapong[1], Phichayut Phinyo[2,3]*, Jayanton Patumanond[2], Sanguansin Ratanalert[4], Kriengsak Limpastan[1]

1 Department of Surgery, Faculty of Medicine, Chiang Mai University, Chiang Mai, Thailand, 2 Department of Family Medicine, Faculty of Medicine, Chiang Mai University, Chiang Mai, Thailand, 3 Center for Clinical Epidemiology and Clinical Statistics, Faculty of Medicine, Chiang Mai University, Chiang Mai, Thailand, 4 Faculty of Medicine, Prince of Songkla University, Songkhla, Thailand

* phichayutphinyo@gmail.com

## Abstract

### Background

The majority of clinical decision rules for prediction of intracranial injury in patients with mild traumatic brain injury (TBI) were developed from high-income countries. The application of these rules in low or middle-income countries, where the primary mechanism of injury was traffic accidents, is questionable.

### Methods

We developed two practical decision rules from a secondary analysis of a multicenter, prospective cohort of 1,164 patients with mild TBI who visited the emergency departments from 2013 to 2016. The clinical endpoints were the presence of any intracranial injury on CT scans and the requirement of neurosurgical interventions within seven days of onset.

### Results

Thirteen predictors were included in both models, which were age ≥60 years, dangerous mechanism of injury, diffuse headache, vomiting >2 episodes, loss of consciousness, post-traumatic amnesia, posttraumatic seizure, history of anticoagulant use, presence of neurological deficits, significant wound at the scalp, signs of skull base fracture, palpable stepping at the skull, and GCS <15 at 2 hours. For the model-based score, the area under the receiver operating characteristic curve (AuROC) was 0.85 (95%CI 0.82, 0.87) for positive CT results and 0.87 (95%CI 0.83, 0.91) for requirement of neurosurgical intervention. For the clinical-based score, the AuROC for positive CT results and requirement of neurosurgical intervention was 0.82 (95%CI 0.79, 0.85) and 0.84 (95%CI 0.80, 0.88), respectively.

**Funding:** The author(s) received no specific funding for this work.

**Competing interests:** The authors have declared that no competing interests exist.

## Conclusions

The models delivered good calibration and excellent discrimination both in the development and internal validation cohort. These rules can be used as assisting tools in risk stratification of patients with mild TBI to be sent for CT scans or admitted for clinical observation.

## Introduction

Mild traumatic brain injury or mild TBI is one of the most common injuries encountered in the emergency departments with the reported incidence ranging from 100–700 per 100,000 person-year [1]. The definition of mild TBI is primarily based on the Glasgow Coma Scale (GCS) score of 13–15 following a blunt head trauma. This was in fact an oversimplification as the severity and prognostic spectrum of mild TBI varies according to numerous relevant factors, which raises substantial burdens to clinicians' management decisions. Only a small proportion, 15–30%, of patients with mild TBI was found to have intracranial injury from computed tomography scan (CT scan) of the brain, and only a few, roughly 1%, required neurosurgical intervention in the end [2]. Therefore, not all patients with mild TBI should undergo CT scans or even be admitted for neurological observations, but can be safely discharged from hospital with appropriate follow-up visits [3]. A proper and accurate decision making would result in a more effective chain of patient cares, a reduction in number of unnecessary CT scans or referral costs, and an improvement in patient outcomes [3–5].

Over the past decades, several clinical decision rules for prediction of the presence of intracranial injuries from CT scans or the requirements of neurosurgical interventions have been developed to reduce the uncertainty during decision making and promote a more selective investigations or patient referrals. The Canadian CT head rules (CCHR) and the New Orleans Criteria (NOC) are among the most widely used and cited tools for risk stratification of patients with mild TBI [3, 6, 7]. However, the majority of the prediction rules were developed from western or high-income countries, where the cause of injury was commonly due to falling. The implication of these rules in low or middle-income countries, where traumatic injury were the results of traffic accidents, is questionable. The different mechanisms of injury also reflect the dissimilarities in other clinical characteristics of the patients such as age distribution. Moreover, the definition of mild TBI and the inclusion criteria was inconsistent across literature. In the case of the CCHR, only mild TBI patients with history of loss of consciousness, amnesia, or disorientation were included, whereas patients with minor injuries, coagulopathy, posttraumatic seizure, and focal neurological deficits were excluded from the development cohort, which strongly limits the generalizability of the CCHR to our setting [3, 7].

This study aims to develop and internally validate the highly sensitive multivariable clinical decision rules for prediction of both the presence of intracranial injury and the requirements of neurosurgical interventions in patients with mild TBI who visited the emergency departments in low-to-middle income countries.

## Material and methods

### Study design and patient cohort

This development and internal validation of multivariable clinical prediction rules was based on a secondary analysis of our recently published, prospective cohort data. This cohort

involved two large medical centers in Chiang Mai: Chiang Mai University Hospital (CMU), and Nakornping Hospital (NKP).

Patients with mild TBI visiting the two centers from December 1st, 2013 to January 31st, 2016 were assessed for eligibility. The inclusion criteria were patients with a history of blunt head injury, aged ≥16 years, and Glasgow Coma Scale (GCS) 13–14 or GCS 15 with one of the following signs or symptoms: diffuse headache, vomiting, loss of consciousness, posttraumatic amnesia, posttraumatic seizure, drug or alcohol intoxication, history of previous neurological procedure, current anticoagulant user (except antiplatelet), signs of skull base fracture, palpable stepping at the skull, and significant wound at the scalp. Patients with uncertain history of trauma and time from onset of injuries more than 24 hours were excluded.

Eligible patients were evaluated and managed according to the local mild traumatic brain injury guidelines. Patients who were highly suspicious of having intracranial injury would be sent for an emergency CT scan of the brain and be treated accordingly. Patients with indefinite signs of intracranial injury might be admit to an observational unit for at least 24 hours from the onset of injury. If any clinical deterioration occurred during the observation period, the patient would be sent for an emergency CT scan. For patients with stable condition without signs of intracranial injury, the doctor might decide to discharge the patient from hospital with an appointment to a follow up visit. The choice of investigation and treatment was left to the discretion of the attending emergency physicians. As CT scan might not be done to verify the occurrence of clinical endpoint in all patients, a prognostic criterion was set up by arranging a clinical follow up visit at 7 days from injury. Patients whose signs/symptoms were not improved, or progressed, or could not go to regular work were scheduled for CT scans. Patients who were not present to the visit were contacted by telephone for an assessment of their conditions by our research staff.

The institutional review boards of the Faculty of Medicine of Chiang Mai University and the Nakornping Hospital approved this study. All data were prospectively collected in a standardized case record form in both participating centers. As patients were routinely managed by local staffs without interference from the study protocol, the informed consent was waived. The reporting of the study was in compliant with the Transparent Reporting of a Multivariable Prediction Model for Individual Prognosis or Diagnosis statement (TRIPOD).

## Potential predictors

Thirteen clinical indicators were selected as potential predictors for positive CT findings based on previously reported literature, expert consensus and local traumatic brain injury guidelines. These comprised of age ≥60 years, dangerous mechanism of injury (defined as motorcycling or cycling without protective helmet, being thrown out of a vehicle, being hit by a motor vehicle, and falling from heights greater than 1 meter), the presence of diffuse headache, vomiting >2 episodes, loss of consciousness, posttraumatic amnesia, posttraumatic seizure, history of anticoagulant use (excluding antiplatelet such as Aspirin), presence of focal neurological deficits, significant wound at the scalp, signs of skull base fracture, palpable stepping at the skull, and GCS <15 at 2 hour follow up. All of the predictors were dichotomous variable except for the patient's age. The age was split into two categories at the cut-point of 60 years old, which was based on the local classification of elderly patient. Due to the advantage of prospective design, an evaluation of all predictors was blinded to the clinical endpoints to be predicted. In our practice, alcohol intoxication is a clinical suspicion by the attending physicians (the patients or witnesses reported alcohol drinking before injuries or the smell of alcohol was detected from the patients' breath). As blood alcohol concentration was not routinely measured in all patients to validly confirm the presence of alcohol intoxication, its reliability as a

predictor within the model is low. Thus, we did not include alcohol intoxication within our models.

## Clinical endpoints

The primary clinical endpoint for the model prediction was the presence of any traumatic intracranial finding on the CT scan of the brain, which included any types of intracranial hemorrhage (e.g. subdural hemorrhage, epidural hematoma, subarachnoid hemorrhage and intracerebral hematoma), and depressed skull fracture. Linear skull fracture was not considered as our intracranial findings of interest. The unblinded neuroradiologists interpreted and reported the official CT results as routinely done in practice.

The secondary clinical endpoint was the requirement of any neurosurgical interventions within seven day from the onset of injury. The neurological procedures included craniotomy or craniectomy, elevation of skull fracture, external ventricular drainage, Burr holes and intracranial pressure monitoring.

## Study size estimation

As there is no consensus for the standard methods for calculation of the adequate study size for the multivariable prediction model, we based our estimation of the study size on the events per variable (EPV) concept. The rule requires at least ten clinical endpoints per one predictor variable. Thus, to adequately power our study with 13 potential predictors, a total of 130 clinical events were required. The sample size, which exceeds the EPV would also help in the prevention of model overfitting.

## Statistical analysis

All statistical analyses were performed using Stata statistical software version 16.0 (StataCorp, Texas). We used frequency and percentage for description of categorical data. For continuous data, descriptive statistics were reported according to the data distribution. The comparison among two independent groups was done with exact probability test for categorical data. Independent t-test or Mann-Whitney U test was used for comparison of continuous data as appropriate. A two-tailed p-value $<0.05$ was considered as statistically significant. In the presence of significant missing data on clinical predictors, multiple imputations would be done.

## Development of the multivariable prediction scores

We developed two simple prediction score via two different methods. For the model-based scoring system, all predictors were included in the multivariable logistic regression model regardless of their statistical significance from the univariable analysis. Prior to statistical modeling, the presence of collinearity among all predictor variables was assessed using variance inflation factors (VIF). To create a risk score, logit coefficients of each predictor variable were divided by the smallest coefficients, and were subsequently rounded to the closest integer. As some clinically-relevant predictors might be found to be non-significant in the multivariable logistic, we derived another prediction model based on both the results from the model-based score and the consensus of clinical experts at our institutes. For the clinical-based scoring system, all predictors were re-classified into three risk classifications (high risk features, intermediate risk features, and low risk features). The weighted score would be assigned differently to each risk category. It is important to note that only the primary clinical endpoint, the presence of positive CT findings, was regressed on all included predictor variables to derive the model-based score. To ensure the accuracy of the scores in prediction of another endpoint,

neurosurgical intervention, measures of performances would be evaluated and reported for both clinical endpoints.

## Measure of performances

The predictive performance of both the model-based and the clinical-based scoring system was examined in terms of discriminative ability and calibration. For discrimination, the concordance statistics or C-statistics was used. When logistic regression was used for model derivation, the C-statistics was represented with the area under receiver operating characteristic curves or AuROC. The AuROC reflects the model's ability to correctly classify those with and without the clinical endpoint of interest from each other based on the model's predicted probabilities. According to Hosmer and Lemeshow, an AuROC of 0.70–0.80, 0.80–0.90, and above 0.90 was considered acceptable, excellent, and outstanding, respectively [8]. The model calibration was assessed statistically and graphically. The calibration of the prediction model refers to the agreement between the model predictions and the observed, or the actual risk, of clinical outcomes. Hosmer-Lemeshow goodness-of-fit test was used. The model was considered statistically fit if the P-value was more than 0.1. The calibration plot was also used for the visual inspection of the agreement between the derived scores (which were converted from the predicted probabilities) and the observed risk of positive CT findings and requirement of neurosurgical interventions within the derivation cohort.

## Internal validation of the prediction score

Internal validation was performed using bootstrap re-sampling techniques with 500 replicates. The bootstrap procedure involves the whole modeling process and was done only for the model-based scoring system. The apparent model performance, the test model performance and the model optimism were estimated and reported for both clinical endpoints.

## Identifying cut-points for clinical implication

For each model, we proposed two cut-points for clinical applicability. The first cut-point, which was based on the prediction of CT findings, would reflect the need for patient admission to the observational unit. The second cut-point, which was based on the prediction of neurosurgical intervention needs, would reflect the need for an emergency CT scan at the emergency department. To identify the appropriate cut-point, we inspected the sensitivity and the specificity of all possible cut-points of both scoring systems.

## Comparative validation with the CCHR

After the appropriate cut-points were defined, a comparative validation with the widely-accepted Canadian CT Head Rule was done. For the CCHR, an emergency CT scan is required when any one of the following findings was identified in the particular patient with mild TBI: GCS <15 at 2 hours after injury, suspected open or depressed skull fracture, any sign of basal skull fracture, vomiting ≥2 episodes, age ≥65 years, amnesia before impact ≥30 minutes, and dangerous mechanism of injury (pedestrian struck by vehicle, occupant ejected from motor vehicle, and fall from elevation ≥3 feet). Sensitivity, specificity and CT ordering proportions were calculated for the CCHR, and were compared to our newly derived model thereafter.

## Sensitivity analysis

As there were significant differences between the two participating centers in terms of CT ordering proportions (CMU 16.2% vs. NKP 76.0%, p<0.001), rate of positive CT scan results

interpreted by the radiologist (CMU 4.8% vs. NKP 42.5%, p<0.001), and rate of neurosurgical intervention done by the neurosurgeon (CMU 0.6% vs. NKP 10.6%, p<0.001), a sensitivity analysis to address the robustness of the primary statistical model via standard logistic regression was performed using multivariable multi-level logistic regression. In this analysis, the study sites were included as the 2nd level predictors. We compared the model discriminative ability, before and after score transformation, with the use of AuROC from both analyses.

## Results

### The derivation cohort

A total of 1,164 patients with mTBI were included in a recently published prospective, multi-center cohort from 2013 to 2014. In this study, the CT scan was done in almost half of the patients (n = 487, 41.9%). Of these patients, 458 received the CT scan during their initial evaluation in the emergency department, and the remaining 29 were sent for CT scan either during their admissions or at follow up. At the end of the study, there were 244 (21.0%) patients with positive CT findings and 920 (79.0%) patients with negative CT findings (Fig 1). The analysis was done on complete-case basis, as there were no missing data on important predictors for statistical modeling.

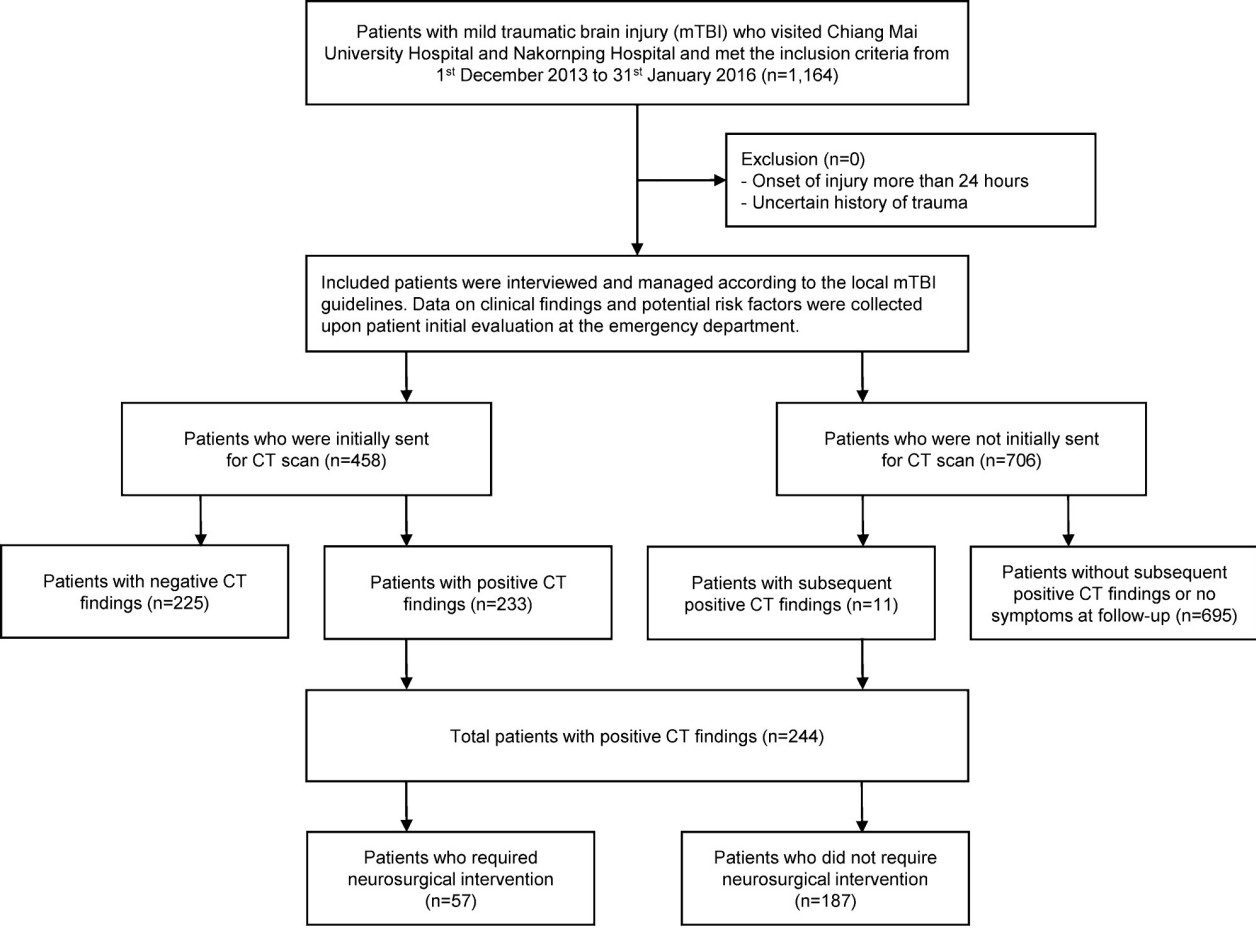

**Fig 1. Patient flow diagram.**

The comparison of the differences in clinical characteristics, mechanism of injury, and clinical predictors between a group of patients with positive and negative CT findings was shown in Table 1. In short, most of the patients were male (65.6%) with a median age of 34 years (IQR 22, 56). Traffic accidents were the most common mechanism of injury (63.4%), followed by falling (24.9%) and physical assaults (9.1%). The detailed results regarding road traffic injuries and other mechanism of injuries are presented in S1 Table.

Several clinical predictors were found to be significantly different among groups such as the presence of diffuse headache, vomiting >2 episodes, posttraumatic amnesia, neurological deficit, significant wound at the scalp, signs of skull base fracture, skull stepping and, GCS <15 at 2 hours. In patients with positive CT scans, common findings were subdural hemorrhage (36.9%), subarachnoid hemorrhage (20.9%), and epidural hematoma (20.5%) (Table 2). Neurosurgical intervention was done in 57 (4.9%) patients. There were no loss to follow up at one week.

**Table 1. Clinical characteristics and predictor variables of the cohort.**

| | Intracranial injuries on CT scan | | | | P-value |
|---|---|---|---|---|---|
| | Present (n = 244) | | Absent (n = 920) | | |
| | n | (%) | n | (%) | |
| **Clinical characteristics** | | | | | |
| Age (years), median (IQR) | 36.5 | (26,54) | 33 | (22,58) | 0.127 |
| Age ≥60 years | 42 | (17.2) | 220 | (23.9) | 0.025 |
| Male | 171 | (70.1) | 593 | (64.5) | 0.111 |
| **GCS on admission** | | | | | <0.001 |
| 15 | 190 | (77.9) | 852 | (92.6) | |
| 14 | 40 | (16.4) | 65 | (7.1) | |
| 13 | 14 | (5.7) | 3 | (0.3) | |
| **Mechanism of injury** | | | | | 0.155 |
| Traffic accidents | 169 | (69.3) | 569 | (61.8) | |
| Falling from height | 48 | (19.7) | 242 | (26.3) | |
| Physical assaults | 21 | (8.6) | 85 | (9.2) | |
| Others | 6 | (2.5) | 24 | (2.6) | |
| Dangerous mechanisms | 15 | (6.2) | 20 | (2.2) | 0.003 |
| Suspected of alcohol intoxication | 73 | (30.0) | 326 | (35.0) | 0.110 |
| **Clinical predictors** | | | | | |
| Diffuse headache | 171 | (70.1) | 206 | (22.4) | <0.001 |
| Vomiting >2 episodes | 37 | (15.2) | 56 | (6.1) | <0.001 |
| Loss of consciousness | 161 | (66.0) | 581 | (63.2) | 0.454 |
| Posttraumatic amnesia | 153 | (62.7) | 508 | (55.2) | 0.042 |
| Posttraumatic seizure | 15 | (6.2) | 35 | (3.8) | 0.112 |
| Current anticoagulant use | 4 | (1.6) | 20 | (2.2) | 0.801 |
| Presence of neurological deficits | 7 | (2.9) | 2 | (0.2) | <0.001 |
| Significant wound at the scalp | 190 | (77.9) | 430 | (46.7) | <0.001 |
| Signs of skull base fracture | 50 | (20.5) | 31 | (3.4) | <0.001 |
| Stepping of the skull | 23 | (9.4) | 17 | (1.9) | <0.001 |
| GCS <15 (at 2 hours) | 45 | (18.4) | 33 | (3.6) | <0.001 |
| **Clinical endpoint (within 7 days)** | | | | | |
| Required neurosurgical intervention | 57 | (23.4) | 0 | (0) | <0.001 |

Abbreviations: CT, computed tomography; IQR, interquartile range; GCS, Glasgow Coma Scale

**Table 2. Intracranial findings from CT scan and types of neurosurgical intervention required.**

|  | n | (%) |
|---|---|---|
| **Intracranial findings from CT brain** |  |  |
| Total number of CT brain | 487 | (41.9) |
| Normal findings | 243 | (20.9) |
| Abnormal findings | 244 | (21.0) |
| Subdural hemorrhage (SDH) | 90 | (7.7) |
| Subarachnoid hemorrhage (SAH) | 51 | (4.4) |
| Epidural hemorrhage | 50 | (4.3) |
| Intracranial hemorrhage (ICH) | 42 | (3.6) |
| Depressed skull fracture | 11 | (1.0) |
| **Types of neurosurgical intervention required** |  |  |
| Total number of neurosurgical interventions (within 7 days) | 57 | (4.9) |
| Craniotomy or craniectomy | 51 | (4.4) |
| Elevate skull fracture | 5 | (0.4) |
| Burr hole | 1 | (0.1) |

## Development of the model-based prediction score

We developed two simple prediction scores via two different methods as described. For the model-based scoring system, the multivariable odds ratios with their 95% confidence intervals, the p-values, the beta coefficients, and the transformed weighted score were listed in Table 3.

**Table 3. Model-based scoring system via multivariable logistic regression model for prediction of positive intracranial injuries from CT scan in patients with mild traumatic brain injuries.**

| Predictors | OR | 95%CI | p-value | ß | Score |
|---|---|---|---|---|---|
| Presence of neurological deficits | 18.3 | 2.6,126.3 | 0.003 | 2.9 | 29 |
| Diffuse headache | 5.8 | 4.0,8.2 | <0.001 | 1.8 | 18 |
| Signs of skull base fracture | 4.8 | 2.8,8.5 | <0.001 | 1.6 | 16 |
| GCS <15 (at 2 hours) | 4.4 | 2.5,7.8 | <0.001 | 1.5 | 15 |
| Dangerous mechanisms | 4.1 | 1.7,10.0 | <0.001 | 1.4 | 14 |
| Stepping of the skull | 3.8 | 1.8,7.9 | <0.001 | 1.3 | 13 |
| Significant wound at the scalp | 2.7 | 1.9,4.0 | <0.001 | 1.0 | 10 |
| Vomiting >2 episodes | 2.0 | 1.2,3.3 | 0.013 | 0.7 | 7 |
| Current anticoagulant use | 1.6 | 0.5,5.5 | 0.451 | 0.5 | 5 |
| Posttraumatic amnesia | 1.3 | 0.9,1.9 | 0.237 | 0.2 | 2 |
| Age ≥60 years | 1.3 | 0.8,2.0 | 0.357 | 0.2 | 2 |
| Posttraumatic seizure | 1.1 | 0.5,2.2 | 0.902 | 0.1 | 1 |
| Loss of consciousness | 1.1 | 0.7,1.5 | 0.912 | 0.1 | 1 |
| Model Intercept | 0.03 | 0.02,0.05 | <0.001 | -3.48 |  |
| Total score |  |  |  |  | 133 |
|  |  | Mean±SE | Min | Max |  |
| Patients with positive CT findings |  | 32.8±1.0 | 0 | 62 |  |
| Patients with negative CT findings |  | 13.2±0.4 | 0 | 79 |  |
| Patients who required surgical intervention |  | 39.9±2.1 | 7 | 79 |  |
| Patients who did not require surgical intervention |  | 16.1±0.4 | 0 | 69 |  |

Abbreviations: OR, odds ratio; CI, confidence interval; ß, beta-coefficient; GCS, Glasgow Coma Scale; SE, standard error; Min, minimum; Max, maximum; CT, computed tomography.

Of thirteen included predictors, eight were found to be statistically significant. The scores, which range from 1 point to the maximum at 29 points, sum up to the total of 133. The presence of neurological deficit, diffuse headache, and signs of skull base fracture were the top three predictors with the highest score. The concordance statistics (C-statistics) via the area under the receiver operating characteristic curve (AuROC) was 0.85 (95%CI 0.82, 0.87) for discrimination of patients based on CT results, and 0.87 (95%CI 0.83, 0.91) for discrimination of patients based on their requirement of neurosurgical intervention (Fig 2A and 2B). For the model calibration, the Hosmer-Lemeshow goodness of fit statistics was insignificant for both outcomes (p-value 0.116 and 0.312, respectively). Calibration plot visualizing the agreement

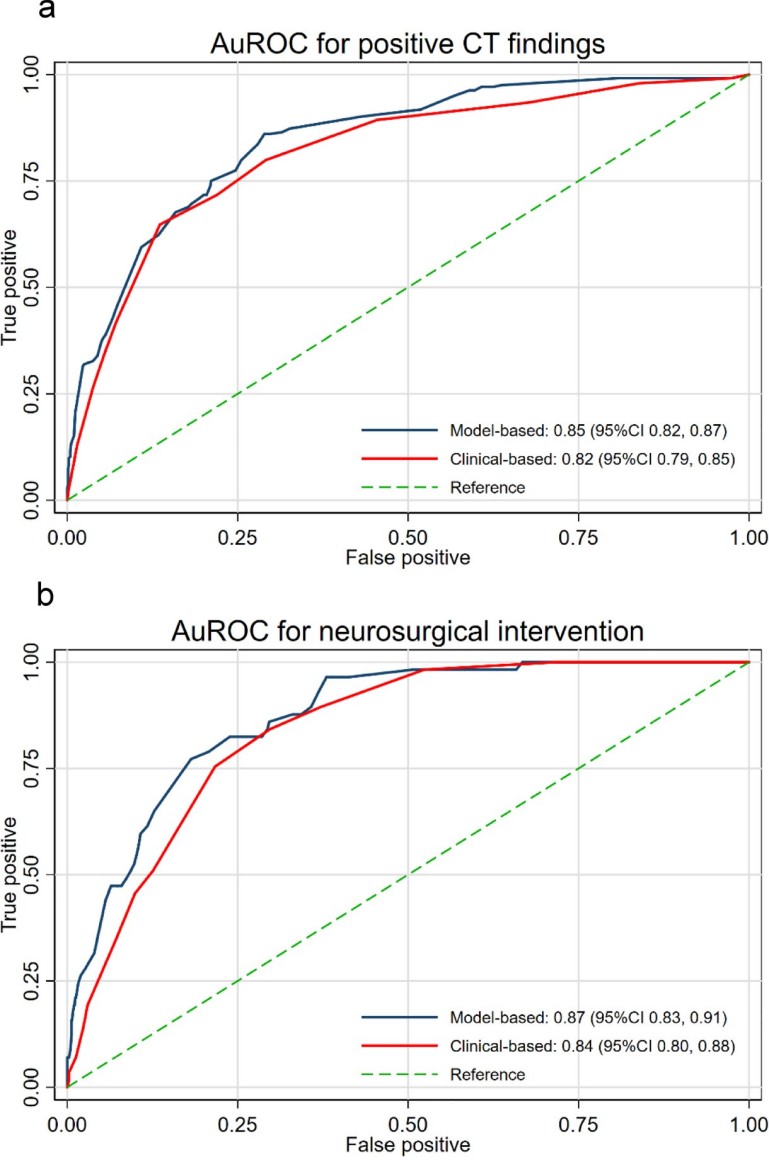

**Fig 2.** Comparison of discriminative ability between the model-based scoring system and the clinical-based scoring system in distinguishing the presence of positive CT findings (2a) and the requirement of neurosurgical intervention within seven days (2b). Abbreviations: AuROC, area under receiver operating characteristic curve; CT, computed tomography; CI, confidence interval.

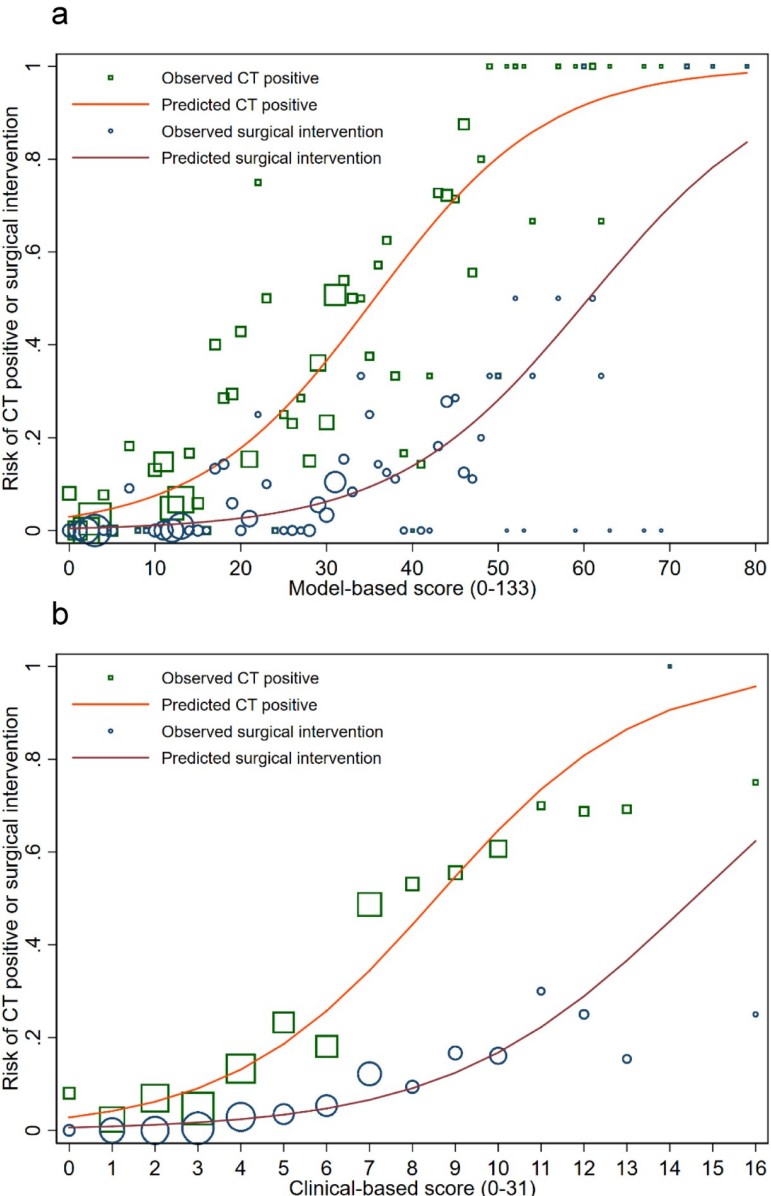

**Fig 3.** Calibration plots visualizing agreement between both the model-based scoring system (3a), the clinical-based scoring system (3b) and the observed risk of positive CT findings or the risk or neurosurgical intervention. The orange line depicts the predicted risk of positive CT findings. The green square represents the observed risk of positive CT findings. The red line depicts the predicted risk of requiring neurosurgical intervention. The blue circle represents the observed risk of requiring neurosurgical intervention.

between the model-based scores and the observed risk of positive CT findings and requiring neurosurgical intervention showed good calibration (Fig 3A).

The C-statistics from bootstrap resampling was 0.8450 (95%CI 0.8448, 0.8452) for discrimination based the on CT results and 0.8714 (95%CI 0.8711, 0.8717) for discrimination based on the neurosurgical requirement. The estimated optimism of C-statistics for both outcomes was 0.0059 and 0.0030, respectively. We presented the detailed results of bootstrap internal validation on S2 Table.

**Table 4. Clinical-based prediction score for prediction of positive intracranial injuries from CT scan in patients with mild traumatic brain injuries.**

| High clinical risk | Intermediate clinical risk | Low clinical risk |
|---|---|---|
| (3 points for each item) | (2 points for each item) | (1 point for each item) |
| Presence of neurological deficits | Dangerous mechanisms | Significant wound at the scalp |
| Diffuse headache | Posttraumatic amnesia | Age $\geq$60 years |
| Signs of skull base fracture | | Loss of consciousness |
| GCS <15 (at 2 hours) | | |
| Stepping of the skull | | |
| Vomiting >2 episodes | | |
| Current anticoagulant use | | |
| Posttraumatic seizure | | |

Abbreviations: GCS, Glasgow Coma Scale.

## Development of the clinical-based prediction score

For the clinical-based scoring system, all thirteen predictors were categorized into three risk categories, which were based on both the result of the model-based multivariable logistic regression and the consensus of clinical experts at our institute. In Table 4, the three clinical risk categories (high-risk features, intermediate-risk features, and the low-risk features) and the predictors within each category are listed. Three points were assigned to the presence of any item within the high-risk category, two points for any item in the intermediate-risk category, and one point for any item in the low-risk category). The clinical-based scores sum up to a total of 31 points. The C-statistics for the clinical-based score was 0.82 (95%CI 0.79, 0.85) for discrimination of patients based on their CT results and 0.84 (95%CI 0.80, 0.88) for discrimination of patients based on their requirement of neurosurgical intervention (Fig 2A and 2B). The Hosmer-Lemeshow goodness-of-fit test was significant for both outcomes (p-value 0.004 and 0.050, respectively). However, a good agreement between the clinical-based score and the observed risk of both outcomes was shown from the calibration plot (Fig 3B).

## Identification of score cut-points and comparative validation

A near to perfect sensitivity of both cut-points was crucial to avoid the occurrence of false-negative cases. For the model-based score, the cut-points were chosen at $\geq$2 (sensitivity 99.2% and specificity 8.2%) for clinical observation, and at $\geq$7 (sensitivity 100% and specificity 33.2%) for emergency CT scan. For the clinical-based model, the cut-points were chosen at $\geq$2 (sensitivity 98.0% and specificity 16.2%) for clinical observation, and at $\geq$3 (sensitivity 100% and specificity 28.3%) for an emergent CT scan (Table 5). With these cut-points, both scoring systems were able to include all patients who finally required neurosurgical intervention to either immediate CT scans or admissions to the observational unit. In contrast, the presence of any risk feature in the CCHR led to an increased in CT ordering proportions compared to the newly-derived decision rules without being able to identify all cases that require neurosurgical intervention.

## Sensitivity analysis

The multi-level logistic regression model where study sites were used at the 2nd level variable exhibited larger effect sizes of each covariate within the model compared to those of standard logistic regression. The discriminative ability of the multi-level model was also higher (AuROC 0.89 (95%CO 0.87, 0.92) vs. 0.85 (95%CI 0.82, 0.87)). However, the AuROC of the

**Table 5. Diagnostic accuracy at each specific cut-point in distinguishing mild TBI patients with positive CT findings from negative CT findings and in distinguishing patients who required neurosurgical intervention from patients who did not.**

| | At & above cut-point | | Below cut-point | | Sensitivity (95%CI) | Specificity (95%CI) |
|---|---|---|---|---|---|---|
| | n | (%) | n | (%) | | |
| **Positive CT findings** | | | | | | |
| Model-based score (cut-point 2) | ≥2 | | <2 | | | |
| Positive CT findings | 242 | (99.2) | 2 | (0.8) | 99.2 | 8.2 |
| Negative CT findings | 845 | (91.9) | 75 | (8.2) | (97.1,99.9) | (6.5,10.1) |
| Clinical-based score (cut-point 2) | ≥2 | | <2 | | | |
| Positive CT findings | 239 | (98.0) | 5 | (2.1) | 98.0 | 16.2 |
| Negative CT findings | 771 | (83.8) | 149 | (16.2) | (95.3,99.3) | (13.9,18.7) |
| Canadian CT Head Rule | Presence | | Absence | | | |
| Positive CT findings | 214 | (87.7) | 30 | (12.3) | 87.7 | 22.7 |
| Negative CT findings | 711 | (77.3) | 209 | (22.7) | (82.9,91.5) | (20.0,25.6) |
| **Surgical intervention required** | | | | | | |
| Model-based score (cut-point 7) | ≥7 | | <7 | | | |
| Surgical intervention required | 57 | (100) | 0 | (0) | 100 | 33.2 |
| No surgical intervention required | 739 | (66.8) | 368 | (33.2) | (93.7,100) | (30.5,36.1) |
| Clinical-based score (cut-point 3) | ≥3 | | <3 | | | |
| Surgical intervention required | 57 | (100) | 0 | (0) | 100 | 28.3 |
| No surgical intervention required | 794 | (71.7) | 313 | (28.3) | (93.7) | (25.6,31.0) |
| Canadian CT Head Rule | Presence | | Absence | | | |
| Surgical intervention required | 54 | (94.7) | 3 | (5.3) | 94.7 | 21.3 |
| No surgical intervention required | 871 | (78.7) | 236 | (21.3) | (85.4,98.9) | (18.9,23.9) |
| **CT ordering proportion** | | | | | | |
| Model-based score (cut-point 7) | 796 | (68.4) | | | | |
| Clinical-based score (cut-point 3) | 851 | (73.1) | | | | |
| Canadian CT Head Rule | 925 | (79.5) | | | | |

Comparison of CT ordering proportions between our newly-derived models and the classic CCHR.

Abbreviations: CI, confidence interval; CT, computed tomography.

multi-level model dropped to 0.85 (95%CI 0.81, 0.87) after score transformation, which was comparable to that of the standard logistic model (S3 Table).

## Discussion

In this study, we have developed and internally validated clinical decision rules for prediction of the presence of intracranial injuries and the requirement of neurosurgical interventions from a secondary analysis of a multicenter prospective cohort of patients with mild TBI who visited the emergency departments in a middle-income country. Thirteen clinical predictors were included in the model, which were age ≥60 years, dangerous mechanism of injury, diffuse headache, vomiting >2 episodes, loss of consciousness, posttraumatic amnesia, posttraumatic seizure, history of anticoagulant use, presence of neurological deficits, significant wound at the scalp, signs of skull base fracture, palpable stepping at the skull, and GCS <15 at 2 hours. We presented two prediction rules, which were derived from two different methods, the model-based from statistical modeling and the clinical-based from clinical consensus. Both scoring systems delivered excellent discrimination and good calibration. However, a higher model performance was shown in the model-based decision rule.

These prediction rules can be used by physicians to stratify patients with mild TBI into three risk groups using two identified cut-points, as shown in Table 5. The first cut-point represents the risk for the presence of intracranial injury on CT scan, whereas the second cut-point represents the risk for neurosurgical intervention within 7 days. First, the patients with scores exceeding both cut-points are considered high risk patients. This group of patients should be sent for emergency CT scans to screen for any intracranial abnormalities that should be managed. The second group with scores ranging between both cut-points is considered as an intermediate risk group. Admission of these patients to a neurological observation unit is suggested. In the presence of neurological deterioration during the observation period, the patient should then be sent for an emergency CT scan. The last group with scores lower than the first cut-point is considered as patients with lowest risk of intracranial abnormalities. These patients could be advised and safely discharged home for self-observation. However, an appropriate clinical follow-up might be required in some patients.

Various clinical prediction rules for identifying appropriate mild TBI patients to be sent for CT scan were developed and validated over the past years. However, there were significant clinical and methodological heterogeneity among studies [7]. As there was no general consensus on the definition of mild TBI patients, authors defined it differently according to their local preferences, which hindered the generalizability of their results to other settings. For instance, CCHR and NOC cannot be theoretically applied to mild TBI patients without loss of consciousness, as this specific group of patients was not included in the derivation cohorts of both rules. Any additional exclusions would undoubtedly lessen the external validity of the model of interest. A large external validation of both the CCHR and the NOC was done in a large prospective multicenter study in the Netherlands [9]. In this study, both rules were applied to all patients with mild TBI regardless of previous restriction criteria to prove the generalizability of both rules to a wider group of patients. The diagnostic accuracy was concluded to be indifferent and valid. However, when the CCHR was applied to our cohort, there was a drop in both sensitivity and specificity compared to the previous report. As a result, the CCHR was unable to identify all cases that required neurosurgical intervention, which would be considered unacceptable.

The predictors included within these prediction rules were also inconsistent across literature. Common risk predictors were focal neurological deficit, signs of basilar skull fracture, signs of skull fracture, loss of consciousness, posttraumatic amnesia, headache, drug or alcohol intoxication, posttraumatic vomiting, posttraumatic seizure, anticoagulant use or coagulopathy, dangerous mechanism of injury, and age older than 60 or 65 years [10–12]. All of these factors have been previously identified as significant indicators of intracranial injury in mild TBI patients. The variation in their effect estimates could be explained by the differences in the patients included and the study size. Our study identified another significant predictor that indicates the presence of intracranial injury, which was the presence of significant scalp wound, either laceration or contusion. This was consistent with past cross-sectional study of mild TBI patients, which concluded the association between scalp lacerations and intracranial injuries [13]. However, the detailed characteristics of the scalp wound to be associated with an increased risk was left undetermined.

Even though most of the decision rules were developed in high-income countries [14], we have identified one study from Thailand, a middle-income country, which developed a clinical predictive score of intracranial hemorrhage in mild TBI patients [15]. The predictors included in the score were the same as in our study, except with fewer number. This result of this study and our previous report also confirmed that the clinical indicators for intracranial injuries in Asian populations were similar to those found in western populations, despite the difference in primary mechanism of injury [16]. However, only patients who received CT scans were

included for development of the score. Without the inclusion of full patient cohort, the estimated score might be biased due to patient selection and partial verification of outcomes [17]. Thus, a new clinical prediction rule which captures an "intended to be investigated" population, which were all patients with mild TBI who presented to the emergency departments, should be established.

Our clinical decision rules were developed according to the current methodological standards and carried several strengths. First, the data of all predictors were prospectively collected in a standardized fashion in both participating centers. This minimizes the risk of incorporation bias and missing data. Second, a broad definition of mild TBI was used without any inappropriate exclusion of patient subgroups. The exclusion was done only to patients with uncertain history of trauma or onset of injury longer than 24 hours. Third, all included clinical predictors were routinely assessed in clinical practice through initial history taking and physical examination. Fourth, our study did not exclude patients without CT results. All patients were verified for the final clinical endpoint via the use of prognostic criteria, the clinical follow-up at seven days after injury. Special effort, by telephone interview, was made to ensure that all included patients were verified. Fifth, the statistical modeling was adequately powered, even at a stricter threshold of 15 events per variable. Higher EPV ratio was necessary when low-prevalence predictors were included in the modeling process [18]. Finally, as clinicians were unlikely to apply the rules in practice if the predictors were not logically related to the endpoints, or were given inappropriate weights from statistical modeling [19], we presented both the model-based and the clinical-based scoring system to settle anticipated arguments regarding the statistical weighing of some predictors.

The CT ordering proportion upon the patient's arrival to the emergency department was 39% (458/1,164), which was relatively low compared to the previous study [3]. This was, in fact, modified by the implication of the local mild head injury guideline, which required attending physicians to evaluate the need for CT scan based on multiple risk features. Thus, the patients would only be sent for CT scan or be referred from the surrounding rural hospitals if only intermediate and high risk features were identified. Although this guideline seemed to be effective in reducing the number of CT scans, 39 more CTs were sent during neurological observation and at follow up visits. Eleven of these subsequent CT were found to have clinically important intracranial injuries. When our scoring systems were applied to this cohort, the number of CT scans would be increased to around 68–73%. This may found to be not cost-effective and some patients might be subjected to unnecessary radiation exposures. However, with this relatively high CT ordering proportion, it was guaranteed that no patient who finally required neurosurgical interventions would be missed since their first visit to the emergency department. On the other hand, when the CCHR was applied to the cohort, even more patients would require initial CT scans. We were unable to compare the accuracy of the CHIP prediction rule in our cohort, as the data on the duration of posttraumatic amnesia beyond 30 minutes was not available.

In terms of limitations, there were some points to be elaborated. First, the reliability of some predictors included in the model was uncertain, such as significant scalp wound. The definition of significant was left solely to the judgment of the attending physicians. We could not summarize from out data the definite characteristics of all the wounds, which were claimed to be significant, as they were not adequately documented. Based on a study by Hamrah et al. and our clinical observation, we suggested that patients with mild TBI with large scalp laceration, which might be defined as the wounds with the width of $\geq 4$ cm or the length of $\geq 7$ cm, should be carefully evaluated as it can be associated with intracranial injuries in at least 20% of the patients [13]. However, these cutoffs were speculative, and further studies are still required to accurately define the wound's character that is significantly associated with an intracranial

injury. Second, a total of thirteen predictors were included in both scoring systems and the score for each item were not memorable. Thus, clinicians might find these models difficult to use and rarely apply them in practice, especially in a hectic environment [20]. However, considering the standard evaluation of patients with mild TBI, all of these factors were usually asked and examined by the physicians, who finally incorporated all the gathered data to classify the risk of significant intracranial injury intuitively. Therefore, to improve the applicability and usability of the derived models, we suggest that these models be further developed into a simplified checklist, or a mobile web application, which could be used in a rapid evaluation. Third, there were marked differences in the proportions of ordering CT scans, positive CT scan results and neurosurgical interventions between both centers, which reflect a significant clinical heterogeneity. To account for this issue, a sensitivity analysis of the model derivation was done via the use of multi-level logistic modeling, where study sites were included as the $2^{nd}$ level predictor. Although the logit coefficients showed some differences, the effect directions of all predictors, and the model discriminative ability in terms of AuROC remained the same. This confirms the robustness of our primary statistical modeling. Finally, even though the newly-derived prediction models performed excellent discriminative ability in the original cohort, the implementation of these models were not suggested until their external validity was proven through a large prospective external validation studies in either different settings or different levels of care.

## Conclusions

In conclusion, two highly sensitive clinical decision rules to predict the presence of intracranial injury or requirement of neurosurgical intervention in patients with mild TBI who visited the emergency departments in a middle-income country were developed and internally validated. These models would serve as effective assisting tools for physicians in the risk stratification of patients with mild TBI to be sent for emergency CT scans, admitted for close neurological observations, or safely discharge home. A further validation study is needed to confirm the generalizability and the transportability of both rules prior to their implication in practice.

## Supporting information

**S1 Table. Detailed results regarding road traffic injuries and other mechanism of injuries.**
(DOCX)

**S2 Table. Detailed results of internal validation with the use of bootstrap resampling procedure.**
(DOCX)

**S3 Table. Sensitivity analysis of the model-based prediction score between standard logistic regression and multi-level logistic regression analysis.**
(DOCX)

**S1 Data.**
(XLSX)

## Acknowledgments

The authors wish to acknowledge significant contributions of all relevant medical and nursing staff at Maharaj Nakorn Chiang Mai Hospital (Chiang Mai University Hospital) and Nakornping Hospital for the data collection and their grateful collaborations during the study period.

## Author Contributions

**Conceptualization:** Tanat Vaniyapong, Phichayut Phinyo, Sanguansin Ratanalert, Kriengsak Limpastan.

**Data curation:** Tanat Vaniyapong, Jayanton Patumanond.

**Formal analysis:** Tanat Vaniyapong, Phichayut Phinyo, Jayanton Patumanond.

**Investigation:** Tanat Vaniyapong, Sanguansin Ratanalert, Kriengsak Limpastan.

**Methodology:** Phichayut Phinyo, Jayanton Patumanond, Kriengsak Limpastan.

**Resources:** Phichayut Phinyo, Kriengsak Limpastan.

**Software:** Phichayut Phinyo.

**Supervision:** Sanguansin Ratanalert, Kriengsak Limpastan.

**Validation:** Jayanton Patumanond.

**Writing – original draft:** Tanat Vaniyapong.

**Writing – review & editing:** Phichayut Phinyo, Jayanton Patumanond, Sanguansin Ratanalert.

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
