## [Decision Letter · Decision Letter 0]

24 Jul 2020

PONE-D-20-17284

Development of clinical decision rules for traumatic intracranial injuries in patients with mild traumatic brain injury in a developing country

PLOS ONE

Dear Dr. Phinyo,

Thank you for submitting your manuscript to PLOS ONE. After careful consideration, we feel that it has merit but does not fully meet PLOS ONE’s publication criteria as it currently stands. Therefore, we invite you to submit a revised version of the manuscript that addresses the points raised during the review process.

We look forward to receiving your revised manuscript.

Kind regards,

Itamar Ashkenazi

Academic Editor

PLOS ONE

Journal Requirements:

Reviewers' comments:

Reviewer's Responses to Questions

**Comments to the Author**

1. Is the manuscript technically sound, and do the data support the conclusions?

Reviewer #1: Yes

Reviewer #2: Partly

Reviewer #3: Yes

2. Has the statistical analysis been performed appropriately and rigorously? 

Reviewer #1: I Don't Know

Reviewer #2: Yes

Reviewer #3: Yes

3. Have the authors made all data underlying the findings in their manuscript fully available?

Reviewer #1: Yes

Reviewer #2: No

Reviewer #3: Yes

4. Is the manuscript presented in an intelligible fashion and written in standard English?

Reviewer #1: Yes

Reviewer #2: Yes

Reviewer #3: Yes

5. Review Comments to the Author

Reviewer #1: Well done study.

Could you please give some more details regarding road traffic injuries ?

Car injuries, motorbike injuries, bike injuries, pedestrian, high velocity, low energy

In line 295:

...in table 4 the three clinical risk categories are listed.... would be better.

Reviewer #2: #Major points

What is the impact of this paper?

The clinical decision tool for adult Mild TBI is considered to be almost established

Although the accuracy may increase as new specific factors are added year by year, the essential value does not change. The goal of the clinical decision rule would be to detect 100% of intracranial injuries requiring surgical treatment and rule out unnecessary CT scans.The authors question whether CCHR and NOC developed in high-income countries should be met in low- and middle-income countries because of different cause of injury.

The CCHR derivation cohort certainly has the highest fall rate, but the second is MVC.Therefore, since the existing screening tools also include MVC information, the rule presented in this paper is not novel.

So what are the clinical decision-making tools needed in low- and middle-income countries? I felt it was unclear.Shouldn't it be simpler and more objectively designed to properly access fewer healthcare resources, rather than reduce inappropriate CT?

The items of the rules presented in this paper are almost the same as those of CCHR. Although it was emphasized that there were many traffic accidents, there were few mentions of the injury mechanism, and the overall purpose was inconsistent.

#Minor points

1. Case of injuries are not shown in detail. Especially the details of the traffic accident are needed.

2. Distribution of GCS at time of admission not shown.In addition, it is good to show the time transition of GCS.

3. The clinical predictor “signifficant wound at the scalp” is subjective and ambiguous.

As mentioned in the discussion, it should be generalized and explained for use in hospitals in other low and middle income countries.

Reviewer #3: 1. Was the influence of ethyl poisoning included or not?

2. In line 33 (abstract), why do potential predictors use the ≥60 years and in the further study design ≥16 years?

3. Using the TRIPOD system is a good choice.

4. Interestingly, the authors acknowledge that the 13-item system is difficult in practice and indeed a limitation, as you mentioned effectively.

6. PLOS authors have the option to publish the peer review history of their article (what does this mean?). If published, this will include your full peer review and any attached files.

Reviewer #1: **Yes: **Hans-Peter Simmen, MD, professor of surgery

Reviewer #2: No

Reviewer #3: No

---

## [Author Response · Author response to Decision Letter 0]

4 Aug 2020

Responses to Reviewers’ comments 

Development of clinical decision rules for traumatic intracranial injuries in patients with mild traumatic brain injury in a developing country 

[PONE-D-20-17284]

Dear Editor and reviewers,

 We would like to thank you for the opportunity to revise our manuscript to be qualified for publication in your journal. We have revised and modified some parts of our manuscript as addressed in response to reviewers’ comments. We hope that our responses and revisions would substantially improve the quality of our manuscript and would be qualified for publication in the journal. If there were any further questions or minor points to be addressed or elaborated, please let us know. We would be more than eager to make any further revision.

Reviewer #1: 

Could you please give some more details regarding road traffic injuries? Car injuries, motorbike injuries, bike injuries, pedestrian, high velocity, low energy

• The details regarding road traffic injuries were added within the results section of the manuscript and the supplementary Table S1 as suggested.

• We found some mistakes regarding the number of mechanism of injuries in Table 1. Thus, we corrected these number accordingly.

Mechanism of injury Intracranial injuries on CT scan Total

(n=1164)

 Present

(n=244) Absent

(n=920) 

 n (%) n % n (%)

Motorcycle accident 

 Did not wear helmet 148 60.7 433 47.1 581 49.9

 Wear helmet 5 2.0 66 7.2 71 6.1

Car accident 0.0 

 Not ejected from vehicle 1 0.4 37 4.0 38 3.3

 Ejected from vehicle 7 2.9 8 0.9 15 1.3

Pedestrians hit by a car 4 1.6 13 1.4 17 1.5

Bicycle hit by a car 0 0.0 1 0.1 1 0.1

Falling from height 0.0 

 <1 meter 27 11.1 170 18.5 197 16.9

 1-3 meters 10 4.1 14 1.5 24 2.1

 > 3 meters 6 2.5 4 0.4 10 0.9

 unknown height 5 2.0 54 5.9 59 5.1

Physical assault 21 8.6 85 9.2 106 9.1

Head struck by an object 4 1.6 14 1.5 18 1.5

Fall from bicycle 4 1.6 11 1.2 15 1.3

Head collision while walking 0 0.0 2 0.2 2 0.2

Unknown mechanism 2 0.8 8 0.9 10 0.9

 

In line 295: ...in table 4 the three clinical risk categories are listed.... would be better.

• Corrected as suggested.

Reviewer #2:

What is the impact of this paper?

• We agreed with the reviewer that our study was not a novel concept and that the supporting evidence on clinical decision tools or predictors of intracranial injuries in adult mild TMI patients is almost established. 

• However, we believed that clinical decision tools should be adapted to local context and situation. The association of reported predictors and the presence of intracranial injury might be affected by the mechanism of injury. As we have stated in the introduction part, most developed prediction rules were from high-income countries where the main mechanism of injury were failing. Thus, there were only a small proportion of patients with motor vehicle accident included. It was evident from out study that the CCHR was unable to identify all cases that required neurosurgical intervention, which would be considered unacceptable.

• We also believed that our newly developed clinical decision rules would have more generalizability to the CCHR and the NOC, as only patients with history of loss of consciousness or amnesia were included (In this study, all patients were included regardless of history of loss of consciousness. We also included patients with coagulopathy, seizure, and skull fracture in our study. 

• We also presented two cut-points for each decision rule. The lower cut-point can be used by physicians to decide whether to discharge the patient or admit the patient for clinical observation. On the other hand, the patients whose score exceed the higher cut-point should be sent for emergency CT scan. This would emphasize the role of clinical observation in the management of adult mild TBI patients and might possibly reduce the number of unnecessary CT scan, which would be appropriate for clinical application in low- and middle-income countries. 

• We already had discussed the clinical use of our models in the discussion part.

#Minor points

1. Case of injuries are not shown in detail. Especially the details of the traffic accident are needed.

• The details regarding road traffic injuries were added within the results section of the manuscript and the supplementary Table S1 as suggested.

• We found some mistakes regarding the number of mechanism of injuries in Table 1. Thus, we corrected these number accordingly.

2. Distribution of GCS at time of admission not shown. In addition, it is good to show the time transition of GCS.

• The distribution of GCS at the time of admission was added to Table 1 as suggested. However, we did not have the data on the time transition of GCS.

3. The clinical predictor “significant wound at the scalp” is subjective and ambiguous.

As mentioned in the discussion, it should be generalized and explained for use in hospitals in other low- and middle-income countries.

• In our practice, the definition of significant scalp wound was left solely to the judgment of the attending physicians. We acknowledge this part in the discussion section.

• There were little evidence concerning the association between characteristic of scalp wounds and the severity or the presence of intracranial injury in literature. 

• We added the following statement as our clinical suggestion in the limitation section:

• Based on a study by Hamrah et al. and our clinical observation, we suggested that patients with mild TBI with large scalp laceration, which might be defined as the wounds with the width of ≥4 cm or the length of ≥7 cm, should be carefully evaluated as it can be associated with intracranial injuries in at least 20% of the patients. However, these cutoffs were speculative, and further studies are still required to accurately define the wound's character that is significantly associated with an intracranial injury.

Reviewer #3: 

Was the influence of ethyl poisoning included or not?

• We added the detail on the “suspected of alcohol intoxication” in the results section of the manuscript.

• We added the following statement within the method section of the manuscript for clarification:

• In our practice, alcohol intoxication is a clinical suspicion by the attending physicians (the patients or witnesses gave on alcohol drinking before the injury or the physician smell alcohol on the breath of the patient). 

• As blood alcohol concentration was not routinely measured in all patients to validly confirm the presence of alcohol intoxication, its reliability as a predictor within the model is low. Thus, we did not include alcohol intoxication within our model. 

• On the other hand, if we did investigate blood alcohol concentration in every patient and include this as a predictor in our model, the model would not be practical and might not be cost-effective. 

In line 33 (abstract), why do potential predictors use the ≥60 years and in the further study design ≥16 years?

• Age ≥60 is one of the candidate predictors of the presence of intracranial injury in adult mild TBI, whereas age ≥16 years is a part of our inclusion criteria of the patients. In this study, we focused on adult mTBI. Thus, we only included patients aged more than 15 years old (≥16) with Glasgow Coma Scale (GCS) 13-14 or GCS 15 with one of the following signs or symptoms: diffuse headache, vomiting, loss of consciousness, posttraumatic amnesia, posttraumatic seizure, drug or alcohol intoxication, history of previous neurological procedure, current anticoagulant user (except antiplatelet), signs of skull base fracture, palpable stepping at the skull, and significant wound at the scalp.

Using the TRIPOD system is a good choice.

• Thank you for your comment. The reporting of the study was in compliant with the Transparent Reporting of a Multivariable Prediction Model for Individual Prognosis or Diagnosis statement (TRIPOD).

Interestingly, the authors acknowledge that the 13-item system is difficult in practice and indeed a limitation, as you mentioned effectively

• Thank you for your comment. We acknowledged this point in the discussion section.

---

## [Decision Letter · Decision Letter 1]

31 Aug 2020

Development of clinical decision rules for traumatic intracranial injuries in patients with mild traumatic brain injury in a developing country

PONE-D-20-17284R1

Dear Dr. Phinyo,

We’re pleased to inform you that your manuscript has been judged scientifically suitable for publication and will be formally accepted for publication once it meets all outstanding technical requirements.

Kind regards,

Itamar Ashkenazi

Academic Editor

PLOS ONE

Additional Editor Comments (optional):

Reviewers' comments:

Reviewer's Responses to Questions

**Comments to the Author**

1. If the authors have adequately addressed your comments raised in a previous round of review and you feel that this manuscript is now acceptable for publication, you may indicate that here to bypass the “Comments to the Author” section, enter your conflict of interest statement in the “Confidential to Editor” section, and submit your "Accept" recommendation.

Reviewer #1: All comments have been addressed

Reviewer #2: All comments have been addressed

2. Is the manuscript technically sound, and do the data support the conclusions?

Reviewer #1: Yes

Reviewer #2: Yes

3. Has the statistical analysis been performed appropriately and rigorously? 

Reviewer #1: I Don't Know

Reviewer #2: Yes

4. Have the authors made all data underlying the findings in their manuscript fully available?

Reviewer #1: Yes

Reviewer #2: Yes

5. Is the manuscript presented in an intelligible fashion and written in standard English?

Reviewer #1: Yes

Reviewer #2: Yes

6. Review Comments to the Author

Reviewer #1: The comments have been addressed.

The paper dos not qualify for a Nobel Prize, however it is well done and interesting.

Reviewer #2: To the authors

Thank you for answering my question sincerely.

All the questions you have pointed out have been solved.

There are no additional improvements to this paper.

7. PLOS authors have the option to publish the peer review history of their article (what does this mean?). If published, this will include your full peer review and any attached files.

Reviewer #1: **Yes: **Hans-Peter Simmen, MD, professor emeritus of surgery

Reviewer #2: No

---

## [Editor Report · Acceptance letter]

10 Sep 2020

PONE-D-20-17284R1 

Development of clinical decision rules for traumatic intracranial injuries in patients with mild traumatic brain injury in a developing country 

Dear Dr. Phinyo:

I'm pleased to inform you that your manuscript has been deemed suitable for publication in PLOS ONE. Congratulations! Your manuscript is now with our production department. 

Kind regards, 

on behalf of

Dr. Itamar Ashkenazi 

Academic Editor

PLOS ONE